# Food Consumption of People with Sickle Cell Anemia in a Middle-Income Country

**DOI:** 10.3390/nu15061478

**Published:** 2023-03-19

**Authors:** Tamara Vilhena Teixeira, Ana Carolina Feldenheimer Da Silva, Cláudia dos Santos Cople Rodrigues, Flávia dos Santos Barbosa Brito, Daniela Silva Canella, Marta Citelli

**Affiliations:** Nutrition Institute, Rio de Janeiro State University, São Francisco Xavier Street, 524, Rio de Janeiro 20550-900, Brazil

**Keywords:** food consumption, NOVA classification, sickle cell anemia

## Abstract

Sickle cell anemia (SCA) is a genetic and hemolytic disease globally characterized by social vulnerability. Food consumption has been insufficiently analyzed in SCA. Secondary iron overload is often observed. This leads to unreliable recommendations for dietary iron restriction. We assessed food consumption and iron intake among adults with SCA. Considering the guidelines for healthy eating, foods were grouped according to the NOVA classification. This transversal study included 74.4% of eligible patients who were registered in the reference center for SCA treatment in Rio de Janeiro, Brazil, in 2019. Data on food consumption were collected through 24 h recall. The monthly household income of 82.3% of patients was less than $770. The consumption of fresh or minimally processed foods was directly associated with monthly household income (*p* < 0.0001; η_2_ = 0.87). Ultra-processed foods provided more than one-third of the total energy intake (35.2%). The prevalence of inadequate iron intake was about 40% among women, while that of iron intake above the tolerable upper limit was 0.8%. People from lower socioeconomic classes had the lowest iron intake. Strategies to encourage the consumption of fresh or minimally processed foods are needed considering the requirement of an antioxidant diet in SCA. These findings highlight the need for health equity to ensure food security and healthy eating in SCA.

## 1. Introduction

Sickle cell disease (SCD) is a multisystemic and hereditary disorder. The global number of individuals with SCD is unknown due to the scarcity of epidemiological data. However, it is estimated that more than 300,000 people are born with the disease each year, the majority in sub-Saharan Africa [1,2,3,4]. Demographic projections suggest that by 2050 this number will exceed 400,000 [5]. In the United States, 1 in 365 African Americans are born with SCD, and the disease is estimated to affect approximately 100,000 Americans [3]. SCD is a prevalent disease in Brazil. Data from the Brazilian Ministry of Health’s Neonatal Screening Program estimate that, in 2019, one in 4340 live newborns in Brazil presented the disease (data provided via email by the technical consultant of the National Neonatal Screening Program on 10 November 2021 at 11:05 am). Currently, there are about 60,000 to 100,000 affected individuals across the country [6].

The homozygous HbSS genotype is associated with the most severe form of SCD, sickle cell anemia (SCA) [1]. SCA is particularly common among individuals with an African ancestry [1,2]. In the United States, 273 African American individuals are born with SCA per 100,000 live births [3] and the poverty rate among the African American population was 18.8% in 2019 [7]. In Brazil, in 2019, a quarter of the total population was considered poor or extremely poor, and 73% of this population was black or brown [8]. Thus, a substantial vulnerability to food and nutritional insecurity is presumed in this population [9,10]. Despite this, the food consumption of people with SCA has been under investigated, and it is unknown whether they are more likely to consume ultra-processed foods (UPF).

SCA is characterized by chronic anemia due to excessive hemolysis [1]. Patients with SCA may experience iron overload (IOL) due to the large number of blood transfusions [11]. The contribution of dietary iron to IOL in patients with SCA is uncertain [12]. Nevertheless, uncertainties regarding food consumption and iron absorption rates lead to unreliable recommendations for dietary iron restriction [13,14]. Excess iron is toxic because it leads to the oxidation of fundamental biological compounds [15]. However, iron deficiency is one of the most common nutritional deficiencies worldwide [16]. To understand the risk of inadequate iron intake in individuals with SCA, it is necessary to assess iron intake, which in turn will help to develop appropriate recommendations that do not induce iron overload or deficiency.

To the best of our knowledge, food consumption and iron intake in a representative sample of homozygous adults with SCA have not been assessed. Therefore, the objective of this study was to assess food consumption and iron intake in adults with SCA, considering the socioeconomic aspects of a middle-income country metropolis population. NOVA classification was considered the best choice to evaluate the diet of people with SCA since this classification assumes that the extent and purpose of the processing to which foods are submitted determine not only their nutrient content, but other attributes with the potential to influence several diseases [17]. Furthermore, an ultra-processed diet is associated with biochemical alterations such as oxidative stress and inflammation [18], which are conditions related to worse prognoses in SCD [19].

## 2. Materials and Methods

### 2.1. Study Population and Location

This a transversal study, carried out in the second largest Brazilian State—Rio de Janeiro—with more than 17 million inhabitants, with one in 1200 newborns estimated to present with the disease. This study included adults with SCA regularly enrolled in the State Institute of Hematology Arthur Siqueira Cavalcante (Hemorio, Rio de Janeiro, Brazil), which is a reference center for SCA patients’ treatment and has data from all patients registered in the state of Rio de Janeiro. On confirmation of SCA diagnosis, patients are enrolled in the Hemorio and called once a year for consultations and examinations and/or whenever the patient has complications. Therefore, patients enrolled in the Hemorio can be considered a representative sample of patients with SCA in Rio de Janeiro. Individuals were recruited between January and December 2019. The inclusion criteria were 19–59 years of age, HbSS genotype, residing in the city of Rio de Janeiro in January 2019, being regularly enrolled in and attending annual activities conducted by the Hemorio. The exclusion criteria were pregnancy and lactation, presence of cognitive impairment, and change in municipality at the time of interview.

### 2.2. Data Collection, Study Variables, and Processing

The questionnaires were submitted to a pre-test in an independent sample of three individuals with sickle cell disease who presented the same socioeconomic characteristics of the studied population. The order, intelligibility, and clarity of the questions were evaluated. After pre-testing, minor changes were made. All individuals that met the inclusion criteria and attended Hemorio in 2019 were invited to participate in the study. Five trained and supervised interviewers conducted the interviews. The patient was asked about food consumption based on an adapted 24 h recall (R24h) method, like that used in a previous study [20]. The application of a single R24h followed the multiple-pass method [21] developed in 1999 by the United States Department of Agriculture. The visual resources of a photographic manual for food quantification were used to improve the recording of food portions reported by the interviewees [22].

Food items were grouped according to the NOVA classification (Group 1. Unprocessed or minimally processed foods; Group 2. Processed culinary ingredients; Group 3. Processed foods; Group 4. Ultra-processed foods), which considers the extent and purpose of industrial food processing [17] to assess total energy, total amount of iron intake and its food sources, and total amount of vitamin C intake and its food sources. Synthesis of the referred items in the 24 h recalls grouped according to the NOVA classification is presented in Appendix A. The adequacy of iron consumption was based on the estimated average requirement (EAR). To estimate the average daily intake of iron, vitamin C, and energy, a table of nutritional composition of foods consumed in Brazil was used [23].

The clinical variables collected were: guidance to restrict the consumption of iron-rich foods, development of pica, multiple blood transfusions, use of iron chelation, and use of alcoholic beverages. Additionally, the following socioeconomic variables were also evaluated: sex, age, self-reported race/skin color, education level, beneficiary of federal financial care programs, and socioeconomic classes based on the Brazilian Economic Classification Criterion and the Brazilian Association of Research Companies (ABEP) (A, B1, B2, C1, C2, D, and E) [24]. The data were entered into EpiData software, version 3.1, with double data entry by different typists. Checks were used to detect and correct errors.

### 2.3. Data Analysis

To describe the socioeconomic profile, individuals were categorized into socioeconomic classes (A, B1, B2, C1, C2, D, and E) established by the ABEP. This classification is defined by a score based on the possession of consumer goods, education level of the family head, and access to public services, such as accessibility to running water at home and paved streets, in which class A is the highest socioeconomic class [24].

Food consumption in the population was described based on the percentage of energy, iron, and vitamin C consumed, distributed among the three food groups—in natura or minimally processed foods and culinary preparations based on these foods, processed foods, and UPF, according to the NOVA classification [17]. Iron intake and its sources were also described according to socioeconomic classification.

The prevalence of inadequate iron intake was calculated using the EAR as a cutoff point, according to the sex and age of the participants [25]. In this method, the prevalence of inadequacy corresponds to the proportion of individuals with an intake below the EAR established for iron. Due to the absence of a normal distribution of the continuous variables in this study, non-parametric tests were used. The relative intake of iron above the upper limit was also described.

Thus, comparisons between two groups were assessed using the Mann–Whitney test, and those among three or more groups were assessed using the Kruskal–Wallis test. The level of significance was set at 5% (*p* < 0.05). A general linear model was used to evaluate the association between socioeconomic variables and food consumption. Analyses were performed using the Statistical Program for Social Sciences (SPSS), version 22.0. A nonparametric test for trends across socioeconomic groups (nptrend) was performed using STATA 16.

### 2.4. Ethical Aspects

This study was approved by the Research Ethics Committee of the Hemorio (number 2464147) and was performed in accordance with the ethical standards of the 1964 Declaration of Helsinki. All patients were previously informed about the study and signed an informed consent form.

## 3. Results

### 3.1. Sociodemographic and Clinical Characteristics of the Participants

The final sample included 215 adult participants, corresponding to 66.2% of the participants enrolled in the Hemorio, who met the inclusion criteria (*n* = 325), and 74.4% of the eligible individuals (*n* = 293) (Figure 1). The mean patient age was 36.6 ± 11.5 years for women and 33 ± 10.6 years for men. Most (61.9%) patients declared themselves black. Approximately 30% of the participants were beneficiaries of federal financial assistance programs (Table 1).

Most participants had already been instructed by health professionals to restrict foods that are considered natural sources of iron. Liver was the main food restricted from the diet, followed by other viscera, red meat, dark green vegetables, and beans. Regarding the risk of IOL, 64.7% of the participants had received >10 blood transfusions during their lifetime. Most (78.6%) participants did not consume alcohol regularly (Table 1). The correlation between iron intake and the occurrence of pica, transfusions, advice to restrict food sources of iron, and use of an iron chelator was tested and no significant association was observed.

### 3.2. Food Consumption According to the NOVA Classification

The average daily energy consumption per capita was 2277.8 ± 1634.9 Kcal, of which 58.4% of food consumption was from in natura or minimally processed foods or from culinary preparations based on these foods, 6.4% was from processed foods, and 35.2% was from UPF (Table 2).

The average per capita intake of iron was 13.2 ± 10.5 mg/day (Table 2). The average iron intake in participants who were instructed to restrict iron intake from food sources was 13.3 ± 11.2 mg/day, whereas that in participants who did not receive iron restriction advice was 14.3 ± 9.5 mg/day. There was no statistically significant difference between these average intakes. More than 72.7% of the total iron intake was from in natura or minimally processed foods and culinary preparations based on these foods, which included mostly beans (23.5%) and meat (31%; such as beef, pork, poultry, and the viscera). Fresh unpackaged bread was the main contributor to iron intake among processed foods, while none of the UPF individually contributed significantly to iron intake. The consumption of foods based on corn and wheat flour, whose fortification is mandatory in Brazil, accounted for 17.5% of the total iron intake (Table 2).

The average per capita vitamin C intake was 84.2 ± 120 mg; 68.8% of vitamin C intake was from in natura or minimally processed foods and culinary preparations based on these foods, 30.6% of vitamin C intake was from UPF, and 0.6% of vitamin C intake was from processed foods. Most (61.4%) participants had a vitamin C intake lower than the EAR. Vitamin C intake from fruits accounted for almost half of the total vitamin C intake (Table 2).

### 3.3. Food Consumption and Iron Intake According to Socioeconomic Classification

Most participants were from socioeconomic classes C + D + E, that is, the household income of 82.3% of patients was less than $770 per month. Consumption of fresh or minimally processed foods was directly associated with monthly family income and the strength of this association was 87% (Table 3). Individuals from classes A + B1 + B2 and C1 had the highest iron intake, while those from classes C2 and D–E, the lowest income classes, had the lowest iron intake (Figure 2A). Iron intake from UPF was higher in the highest income group (Classes A + B1 + B2) than the iron intake from UPF in classes C1, C2, and D–E. Similarly, iron intake from meat was higher in the highest income group than the iron intake from meat in classes C2 and D-E (Table 4).

### 3.4. Iron Intake According to the Advice to Restrict Iron-Rich Foods

The mean iron intake in participants who were advised to restrict foods with high iron content was 13.3 ± 11.2 mg/day, while that in participants who were not advised to restrict iron-rich foods was 14.3 ± 9.5 mg/day; there was no statistically significant difference between the mean intakes. Regarding the main foods that were restricted from the diet, iron intake from any of these foods were not different between individuals who received or did not receive advice to restrict iron-rich foods.

### 3.5. Prevalence of Inadequate Iron Intake

Men had a higher level of iron intake than women did. Among adult women of reproductive age (19–50 years), it was not possible to calculate the percentiles of iron intake since the distribution of iron requirement in this group was asymmetric, as reported in a previous study [26]. However, descriptive analysis of the data revealed that the prevalence of iron inadequacy was approximately 40% in these women. The prevalence of iron inadequacy was 8.0% in women aged >50 years. The proportion of participants with iron intake above the tolerable upper limit of intake was small (Figure 2B).

## 4. Discussion

We investigated the food consumption of adults with SCA. Most studies on SCA are done with children because, while in high-income settings the current life expectancy for patients with SCD is estimated to be between 45 and 55 years of age, in low- and middle-income countries it is believed that most children die before reaching adulthood [5,27,28,29]. This large disparity is emphasized by estimates suggesting that 90% of SCD occurs in low- and middle-income countries, and 90% of children with SCD in these countries die before reaching the age of 5 [29]. This study, conducted in a metropolis of a middle-income country, showed that UPF accounted for more than one-third (35.2%) of the total energy intake from the diet in adult people with SCA, which is much higher than that observed in representative samples of the adult Brazilian population (19.7%) [30] or that reported in the southeastern region of Brazil (22.5%) [31] where the state of Rio de Janeiro is located, or even that reported in studies conducted in other countries [32,33]. The data reported here showed that the consumption of fresh foods was directly influenced by family income. Inadequate food and nutrition intake increased the risk of worse prognoses in SCA [11], and correction of nutrition is critical for improving clinical outcomes in patients with SCA [27].

For the general population, nutritional frailty—a multidimensional intermediate status in the ageing process [34]—is a possible consequence of an imbalance in iron intake and is related to an impairment in the quality of life. A nutritional recommendation for iron intake has not yet been established in SCA. Some studies have shown that serum concentrations of hepcidin are lower in people with SCA without iron overload, which may cause an increase in the intestinal absorption of this nutrient [13]. On the other hand, transfusion iron overload increases hepcidin synthesis [13] and could promote a reduction in the intestinal absorption of iron. Recent studies have shown that, in a murine model of SCA, the reduction in iron intake reduced the occurrence of hemolytic events [35,36]. Therefore, the effect of iron intake on the health of people with SCA is still uncertain and therefore, for analysis purposes, we used the estimated average requirement (EAR) proposed for the general population by the US Institute of Medicine [25]. Most of the iron intake in the present study was from in natura or minimally processed foods. In this group of foods, beans and meat were the main contributors to iron intake. In addition, they were the most reported foods that should be restricted from the diet by 57.2% of the participants. Despite this, there was no difference in the intake of iron between individuals who were counseled to restrict iron and those who were not. Thus, counseling to restrict iron source foods does not seem to be the cause of insufficient iron intake.

In contrast, evaluation of iron intake, considering the different socioeconomic classes, showed that individuals from the highest income group had higher iron intake than those from the lowest income group. These results point to a scenario of nutritional vulnerability that may have clinical consequences. As in other diseases, education level and socioeconomic status directly affect the nutritional status in SCA. There is a structural problem in Brazil and in other countries such as the United States, which is related to racial inequality [37] and implies the worst socioeconomic conditions of the population with SCA [38].

Considering that UPF, beans, and meats are important sources of iron, we investigated their respective contribution to iron intake. In the highest income group, the highest intake of iron was from UPF and meats. The contribution of UPF to the total daily energy consumption increases with increasing income [33], and higher consumption of UPF has been directly associated with iron intake [26].

Iron fortification of foods plays an important role in combating iron-deficiency anemia in many countries. Mandatory fortification of corn and wheat flour in Brazil has increased the average iron intake [26] and has reduced the incidence of anemia in other countries [39,40]. Despite the positive evidence of these public policies, there is no evidence to guarantee their safety in people with SCA. Thus, while trying to correct iron deficiency, this policy could exacerbate IOL. However, the contribution of cornmeal and wheatmeal-based foods accounted for less than one-fifth of the total iron intake, and the prevalence of iron intake above the tolerable upper limit was only 0.8% and 3.4% among women and men, respectively.

Inadequate iron intake was reported by 48% women participating in the present study. Inadequate iron intake in this group was higher than that in the adult female population aged 19–59 years in Brazil (30.4%) [8]. Although it is always recommended to collect repeated R24h per participant because a single R24h can result in an imprecise measure of an individual’s usual intake due to the day-to-day variation in an individual’s diet [41], the 24 h Dietary Recall is commonly used to describe a population’s intake since the typical mean intakes can be estimated with a single administration in a given population [42]. According to Gibson (2005) [43], a single 24 h recall is sufficient to estimate the mean or median nutrient intake. For this reason, the present study chose to calculate the median iron intake using the EAR as a reference point, according to the sex and age of the participants.

Strategies to encourage the consumption of in natura or minimally processed foods are urgently needed considering the benefits of consuming antioxidants in SCA [19]. Furthermore, since excessive iron intake was not observed, mandatory iron fortification seems to not pose a risk for IOL.

## 5. Conclusions

The findings of this study highlight the need for health equity and better clinical and nutritional monitoring of patients with SCA to ensure food and micronutrient security and healthy eating.

## Figures and Tables

**Figure 1 nutrients-15-01478-f001:**
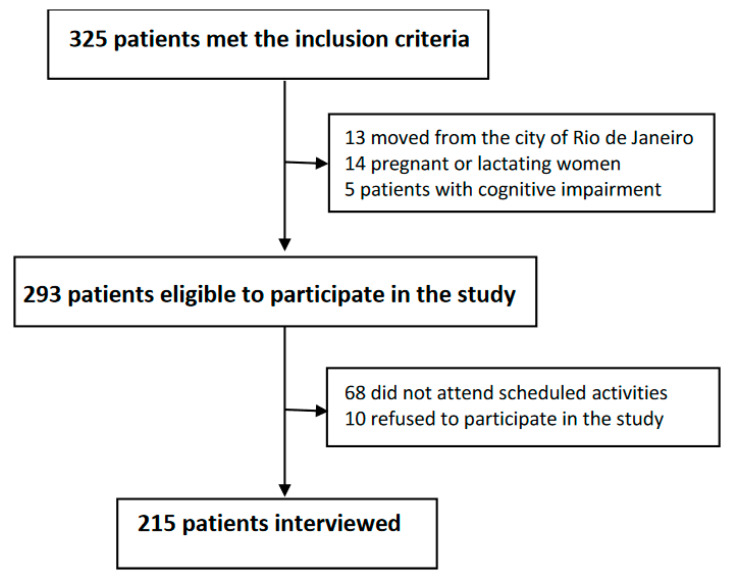
Study flowchart.

**Figure 2 nutrients-15-01478-f002:**
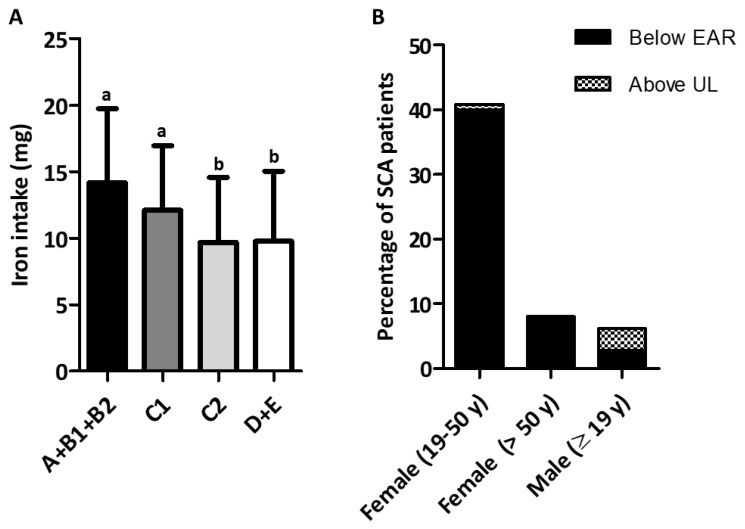
Iron intake of adults with sickle cell anemia. (**A**) Iron intake according to socioeconomic classification. Socioeconomic classes A (*n* = 1) + B1 (*n* = 2) + B2 (*n* = 35); socioeconomic class C1 (*n* = 70); socioeconomic class C2 (*n* = 82); socioeconomic classes D–E (*n* = 25). Values correspond to median iron intake of the respective classes. The same letters indicate no significant difference according to Mann–Whitney test. Nonparametric test for trend across ordered groups (nptrend) was performed (*p* < 0.0001). (**B**) Prevalence of inadequate iron intake and relative intake of iron above upper limit. Estimated average requirement (EAR) for iron: 6 mg (≥19 years); 8.1 mg (19–50 years); 5 mg (>51 years); UL: tolerable upper limit of intake: >45 mg.

**Table 1 nutrients-15-01478-t001:** Socioeconomic and clinical characteristics of the participants (N = 215).

Variables	Frequencies	
Sex	N	%
Female	126	58.6
Male	89	41.4
Age (Years)		
19–30	89	41.4
31–50	98	45.6
51–59	28	13.0
Color/race		
Black	133	61.9
Brown	65	30.2
White	9	4.2
Yellow	6	2.8
Indigenous	2	0.9
Education		
Elementary	65	30.2
Secondary school	124	57.7
Tertiary/University	26	12.1
Benefit from federal financial aid programs		
Yes	58	27.0
Advice to restrict iron source foods *		
Yes	123	57.2
Occurrence of pica		
Yes	40	18.6
Performing multiple blood transfusions **		
Yes	139	64.7
Use of iron chelators		
Yes	16	7.4
Regular alcohol consumption		
Yes	46	21.4

* Main reported foods: liver, viscera, red meat, dark green vegetables, and beans. ** More than 10 transfusions throughout life.

**Table 2 nutrients-15-01478-t002:** Relative and absolute consumption of energy, iron, and vitamin C from foods according to the NOVA classification.

Food Groups and Subgroups	Energy		Iron		Vitamin C	
	kcal/Day (M ± SD) *	% of Total Intake	mg/Day (M ± SD) *	% of Total Intake	mg/Day (M ± SD) *	% of Total Intake
In natura or minimally processed foods **	1330.8 ± 1364.0	58.4	9.6 ± 9.5	72.7	57.9 ± 109.3	68.8
Beef or pork	239.3	10.5	2.5	18.9		
Rice	225.8	9.9	0.2	1.5		
Poultry	185.4	8.1	1.0	7.6		
Beans	142.9	6.3	3.1	23.5		
Other cereals ^a#^	88.6	3.9	0.6	4.5	1.5	1.8
Fruits ^b^	82.9	3.6	0.3	2.3	39.2	46.6
Roots and tubers	82.3	3.6	0.3	2.3	4.6	5.5
Milk	43.0	1.9	-	-		
Eggs	41.6	1.8	0.2	1.5		
Vegetables	39.5	1.7	0.4	3.0	11.2	13.3
Viscera	38.6	1.7	0.6	4.5		
Fish	29.3	1.3	0.1	0.8		
Fermented foods	22.9	1.0	-	-		
Homemade cakes ^#^	20.0	0.9	-	-		
Other in natura or minimally processed foods ^c^	48.8	2.1	0.3	2.3	1.5	1.8
Processed foods	146.1 ± 181.5	6.4	0.5 ± 0.6	3.8	0.5 ± 3.1	0.6
Bread (fresh unpackaged) ^#^	106.6	4.7	0.4	3.1		
Cheeses	20.6	0.9	-	-		
Salted, dried, cured, or smoked meats	10.7	0.5	-	-	0.1	0.1
Fruit in syrup (with or without added antioxidants)	6.8	0.3	-	-	0.4	0.5
Other processed foods ^d^	1.3	0.1	0.1	0.7		
Ultra-processed foods	800.8 ± 849.8	35.2	3.1 ± 4.7	23.5	25.8 ± 52.4	30.6
Carbonated soft drinks and industrialized juices	156.1	6.9	0.2	1.5	18.2	21.5
Ready or semi-ready dishes ^e#^	105.9	4.6	0.8	6.1	0.9	1.1
Cakes, pies, and sweet cookies	95.6	4.2	0.5	3.8	-	-
Sausages and other reconstituted meat products	77.7	3.4	0.4	3.0	-	-
Milk drinks	75.2	3.3	-	-	4.7	5.6
Salty cookies and snack chips ^#^	72.4	3.2	0.3	2.3	1.0	1.2
Margarine	64.7	2.8	-	-	-	-
Breads (packaged) ^#^	51.2	2.2	0.5	3.8	-	-
Candies (confectionery) ^f^	47.8	2.1	0.1	0.8	-	-
Fast food ^g#^	43.3	1.9	0.4	3.0	-	-
Other ultra-processed foods ^h#^	11.2	0.5	0.1	0.8	1.0	1.2
Total	2277.8 ± 1634.9	100.0	13.2 ± 10.5	100.0	84.2 ± 120.0	100.0

** Including culinary preparations based on these foods. ^a^ Corn, oats, wheat, and their flours and noodle dishes. ^b^ Includes fruit and natural fruit juices. ^c^ Butter, sugar, other meats, preparations based on lentils, peas, and soybeans, seafood, coffee and tea, nuts and seeds, oils and olive oil, and cream. ^d^ Canned or bottled vegetables and legumes in brine and canned fish (with or without added preservatives). ^e^ Pizzas, frozen ready-made pasta or meat dishes, instant noodles, and industrialized soups. ^f^ Chocolates, candies, ice creams, and industrialized sweets in general. ^g^ Hamburgers and cheeseburgers, hot-dogs, and fried and baked snacks. ^h^ Ultra-processed cheeses, breakfast cereals, distilled beverages, and industrialized sauces. ^#^ Foods from these groups, made with corn or wheat flour, accounting for 17.5% of the total iron intake. * (M ± SD): mean ± standard deviation.

**Table 3 nutrients-15-01478-t003:** Energy consumption according to socioeconomic classification and monthly household income.

Independent Variable	Dependent Variable Food Groups	F	*p*	Effect Size η^2^
Socioeconomic classes *	*In natura* or minimally processed foods	0.477	0.699	0.007
	Processed foods	1.691	0.170	0.023
	Ultra-processed foods	2.340	0.074	0.032
Monthly household income **	*In natura* or minimally processed foods	5.123	<0.0001	0.870
	Processed foods	0.765	0.917	0.499
	Ultra-processed foods	1.151	0.239	0.600

* Socioeconomic classes A + B1 + B2 (*n* = 38); C1 (*n* = 70); C2 (*n* = 82); D-E (*n* = 25). ** Amount in USD.

**Table 4 nutrients-15-01478-t004:** Items of consumption that contributed most to iron intake and their distribution by socioeconomic classes.

	Classes A + B1 + B2 (*n* = 25)		Class C1 (*n* = 70)		Class C2 (*n* = 82)		Classes D + E (*n* = 25)		*p* Value
Variables	median	IQR	median	IQR	median	IQR	median	IQR	
Ultra-processed foods	4.4a	2.1–8.6	2.1b	1.0–4.5	1.8b	0.9–4.0	2.2b	0.6–4.1	0.004
Meats *	3.1a	0.8–5.1	1.9ac	0.7–6.0	1.3bc	0.0–3.8	1.0b	0.0–2.7	0.041
Bean	2.0	0.0–6.5	2.7	0.0–4.7	2.3	0.0–3.1	2.3	0.0–3.7	0.913

IQR: Interquartile Range (Q1–Q3); * includes beef, pork, poultry, and viscera; *n* = number of individuals; the same letters indicate no significant difference according to Mann–Whitney test; *p* value: descriptive level (significant when *p* ≤ 0.05).

## Data Availability

The data presented in this study are available on request from the corresponding author.

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
