# Peer review of "Food Consumption of People with Sickle Cell Anemia in a Middle-Income Country"

_nutrients, 2023, doi:10.3390/nu15061478_

Round 1

Reviewer 1 Report

In this manuscript, Teixeira and colleagues present data showing estimating the dietary iron intake of patients with sickle cell anemia based on a 24 hr recall questionnaire. Foods and their respective contribution to iron intake are classified by the NOVA method. Data are stratified based on socioeconomic status and sex and iron intake is compared to the estimated average requirement to determine “adequacy”. The authors report that most of the subjects earned less than $770 per month and those from lower socioeconomic classes had lower iron uptake. Inadequate iron uptake was ~40 % for female subjects and less than 1% of the study cohort reported iron intake above the tolerable upper limit. The authors conclude that insufficiency is the greatest concern and suggest strategies are needed to improve consumption of minimally processed food.

 The manuscript may be improved by addressing the following concerns:

1.       Assuming the estimated average requirement of iron is the same in patients with SCA as in the general population is quite a leap. Concentration-function relationships of vitamins and minerals are likely to differ in the disease state. As currently written, the authors assume low iron is “inadequate” but in fact the optimal iron requirement for this population is unknown. A growing body of data suggests that decreasing dietary iron availability may be useful in this setting. The authors might consider including some of that literature in this manuscript and presenting results with a more neutral tone.

2.       Inclusion of data from a reference or control group would be useful in determining whether these observations are specific to SCA patients or simply reflect the regional dietary iron intake in the general population.

3.       It would be useful for the reader to see graphs of the data presented in Tables 3 and 4.

4.       Please define the age groups included in “women of reproductive age” in this study (p8 line 253 and anywhere else it occurs)

5.       Many statements regarding the effects of poor nutrition/improved nutrition in sickle cell anemia cite studies done in children. Because it is unlikely that the impact and effect of altering nutritional status will be the same in an adult population, it is preferable to cite studies performed in adults for these statements.

6.       Several clinical variables were collected but not included in data analysis (e.g., development of pica, transfusions, chelation, etc). It would be interesting to examine the relationship of these variables to iron intake.

Reviewer 2 Report

thanks for involving me in the review of this manuscript

The authors should add as an appendix as supplementary material to see how the foods in the food questionnaire were grouped according to NOVA groups. This is very important to make explicit.

I suggest reviewing the statistical approach by breaking down the analysis by socioeconomic groups and see the effects size on consumption of each food group to see how it changes. It does not make sense to report consumption averages, it is not very informative in terms of synthesizing the findings afterwards.

In Table 3, the authors should run an analysis with p for trend to see how it varies across groups

Line 242: The authors discuss how iron intake is important in terms of nutritional squilirium. In this regard, I suggest mentioning nutritional frailty as a possible consequence of iron intake imbalance, among other things. see 10.1111/joim.13384.

Review then English, please

Round 2

Reviewer 1 Report

The authors have addressed the majority of the concerns of this reviewer.

Reviewer 2 Report

Manuscript well improved